# Complete and Durable Response to Combined Chemo/Radiation Therapy in *EGFR* Wild-Type Lung Adenocarcinoma with Diffuse Brain Metastases

**DOI:** 10.3390/diagnostics9020042

**Published:** 2019-04-11

**Authors:** Davide Adriano Santeufemia, Giuseppe Palmieri, Antonio Cossu, Valli De Re, Laura Caggiari, Mariangela De Zorzi, Milena Casula, Maria Cristina Sini, Giovanni Baldino, Maria Filomena Dedola, Giuseppe Corona, Gianmaria Miolo

**Affiliations:** 1SSD Oncology, Alghero General Hospital, 07041 Alghero, Italy; davidesanteufemia@gmail.com (D.A.S.); giovannibaldino@tiscali.it (G.B.); 2Unit of Cancer Genetics, Institute of Biomolecular Chemistry (ICB), National Research Council (CNR), Traversa La Crucca 3, 07100 Sassari, Italy; gpalmieri@yahoo.com (G.P.); casulam@yahoo.it (M.C.); mc.sini@tiscali.it (M.C.S.); 3Operative Unit of Pathology, Azienda Ospedaliero Universitaria Sassari, Via Matteotti 60, 07100 Sassari, Italy; cossu@uniss.it; 4Immunopathology and Cancer Biomarkers, Centro di Riferimento Oncologico di Aviano (CRO) IRCCS, Via F. Gallini 2, 33081 Aviano, Italy; vdere@cro.it (V.D.R.); lcaggiari@cro.it (L.C.); mdezorzi@cro.it (M.D.Z.); giuseppe.corona@cro.it (G.C.); 5Radiotherapy Unit, Azienda Ospedaliero Universitaria Sassari, Via Matteotti 60, 07100 Sassari, Italy; mdedola@yahoo.it; 6Department of Medical Oncology, Centro di Riferimento Oncologico di Aviano (CRO) IRCCS, Via F. Gallini 2, 33081 Aviano, Italy

**Keywords:** lung cancer, *EGFR*, brain metastases, radiotherapy, chemotherapy, *CDH1*, *TP53*, tumor mutational burden

## Abstract

Most non-small-cell lung cancer (NSCLC) patients are likely to develop brain metastases during the course of their illness. Currently, no consensus on NSCLC patients’ treatment with brain metastasis has been established. Although whole brain radiotherapy prolongs the median survival time of approximately 4 months, a cisplatin-pemetrexed combination may also represent a potential option in the treatment of asymptomatic NSCLC patients with brain metastases. Herein, we report the case of a non-smoker male patient with multiple, large and diffuse brain metastases from an “epidermal growth factor receptor (*EGFR*) wild-type” lung adenocarcinoma who underwent an overly aggressive chemo/radiation therapy. This approach led to a complete and durable remission of the disease and to a long survival of up to 58 months from diagnosis of primary tumor. The uncommon course of this metastatic disease induced us to describe its oncological management and to investigate the molecular features of the tumor.

## 1. Introduction

About 20–50% of non-small-cell lung cancer (NSCLC) patients are likely to develop brain metastases (BMs) during their illness [1,2]. The median survival for untreated NSCLC patients is 2 months once BMs occur [3]. Treatment via whole brain radiotherapy (WBRT) has shown to improve survival time by approximately 4 months after diagnosis and it represents the standard palliative treatment for NSCLC symptomatic patients with multiple BMs who are not candidates to receive either surgery or radiosurgery [4,5,6]. Although the role of chemotherapy remains unclear in this setting, as most chemotherapeutic drugs are not able to cross the blood-brain barrier (BBB) [7], it is thought that a cisplatin-pemetrexed combination may be a viable option in the treatment of asymptomatic epidermal growth factor receptor (*EGFR*) wild-type non-squamous NSCLC patients with BMs [8]. Indeed, it is known that the BBB is disrupted when BMs develop and, in addition, the cerebrospinal fluid penetration of intravenous pemetrexed in patients with BMs has been previously demonstrated [9,10].

Moreover, the administration of four cycles of cisplatin plus pemetrexed, followed, in absence of disease progression, by maintenance of pemetrexed alone, still represents the standard chemotherapy regimen adopted in Italy in fit patients affected by *EGFR* wild-type advanced pulmonary adenocarcinoma without programmed death-ligand 1 (PD-L1) immunohistochemical expression on ≥50% of tumor cells [11,12].

On the other hand, BMs incidence in *EGFR* mutant NSCLC is higher than in wild-type *EGFR* disease [13]. NSCLC patients with activating somatic mutations in the tyrosine kinase domain of *EGFR* gene are highly responsive to tyrosine kinase inhibitors (TKIs), such as erlotinib, gefitinib and afatinib, which are specific for *EGFR* (*EGFR*-TKIs) [13,14,15]. Based on their intracranial activity, TKIs have been shown to be effective for treating BMs, even without the use of radiotherapy upfront. Their employment improved intracranial disease progression free survival (PFS) when compared with non-target chemotherapy despite the development of resistance to these drugs being almost inevitable [13,14,15,16,17].

Unfortunately, *EGFR* wild-type patients with BMs exhibited a significantly poorer response to *EGFR*-TKIs and a shorter survival compared with *EGFR* mutant patients [14,18]. Chemotherapy with cisplatin–pemetrexed is usually planned with only palliative intent for treating *EGFR* wild-type patients with BMs. In this setting it was reported a median overall survival of approximately 11 months in responder asymptomatic patients [8], whereas long-term durable remission is an exceptional event in symptomatic cases with high brain tumor burden [7].

Herein, we present the anomalous case of a non-smoker male patient with multiple, large and diffuse BMs from *EGFR* wild-type lung adenocarcinoma who underwent a combined, highly aggressive chemo/radiation therapy achieving a complete remission of the disease. Indeed, in addition to the WBRT and four cycles of cisplatin-pemetrexed combination, considered the standard treatment [12], we decided to administer two more cycles of the same induction chemotherapy followed by 18 pemetrexed maintenance cycles. Moreover, taking in account the excellent response to previous chemo/radiation therapy and his good performance status on the remaining lung lesion, mediastinal lymph nodes, and residual brain metastasis, a treatment consisting of intensity-modulated radiation therapy (IMRT) and helical tomotherapy (HT) has been performed.

Since the uncommon course of this brain metastatic disease, together with the clinical case description, we saw fit to describe also the molecular intrinsic tumor characteristics. Specifically, we focused our attention on *TP53* and *CDH1* genes which we found to be altered in the tissue sample. *TP53* represents an important tumor suppressor gene, widely known to contribute to oncogenesis of several neoplastic diseases, including lung cancer [19], through transcriptional regulation of numerous genes involved in cell death, cell cycle arrest, senescence, DNA-repair, and many other processes. By contrast, *CDH1* gene codifies for a transmembrane glycoprotein, also known as epithelial cadherin (E-cadherin) that plays an essential role in maintaining cell adhesion and adherent junctions in normal tissues and it is frequently absent in a variety of epithelial tumors, including lung cancer [20], thus promoting cancer invasion and metastasis.

## 2. Case Report

In April 2014, a 62-year-old male, non-smoker came to our attention with a sudden weakness, mainly in right upper limb. A total body computed tomography (CT) scan showed a lesion of 4.5 cm in diameter in the right upper lobe of the lung with an omolateral mediastinal lymphadenopathy and multiple, large and diffuse brain lesions (Figure 1A and Figure 2). After diagnostic bronchoscopy he was histologically diagnosed with lung adenocarcinoma.

*EGFR* mutation analysis (exons 18–21) performed on the biopsy tissue demonstrated no mutation together with no identified concomitant rearrangement of the *ALK* or *ROS1* genes.

The patient was submitted to whole brain radiotherapy (20 Gy delivered in five fractions) and chemotherapy with a schedule consisting of cisplatin (75 mg/m^2^) and pemetrexed (500 mg/m^2^) every 3 weeks. After four cycles of chemotherapy a full body CT scan showed significant shrinkage of primary lung tumor, mediastinal lymph nodes and BMs. Further shrinkage was obtained after six cycles. Subsequently, the patient underwent maintenance chemotherapy with pemetrexed (500 mg/m^2^) administered every 3 weeks. After 18 cycles we decided to stop chemotherapy since the disease had stabilized.

The patient has then been subjected to a restaging via brain magnetic resonance imaging (MRI). This showed a residual metastasis in the left frontal lobe and an 18 fluorodeoxy-D-glucose (FDG) positron-emission tomography/computed tomography (PET-CT) scan revealed an increased fluorodeoxyglucose uptake on the primary tumor site and on the omolateral mediastinal lymph nodes. In January 2016, the patient underwent a final treatment of intensity-modulated radiation therapy (IMRT) and helical tomotherapy (HT) on the remaining lung lesion, mediastinal lymph nodes (60 Gy delivered in 25 fractions), and residual brain metastasis (20 Gy delivered in four fractions). A complete response to IMRT-HT along with signs of fibrosis was observed on follow up imaging (Figure 1B and Figure 3). After 58 months from diagnosis of the primary tumor, the patient continues to enjoy good health with no evidence of disease recurrence.

## 3. Genetic Analyses

DNA was isolated from formalin-fixed paraffin-embedded (FFPE) tissue sections using a standard protocol. Paraffin was removed from FFPE samples by treatment with Bio-Clear (Bio-Optica, Milan, Italy) and DNA was purified using the QIAamp DNA FFPE Tissue Kit (Qiagen Inc., Valencia, CA, USA).

*EGFR* mutation (exons 18–21) was evaluated by pyrosequencing on a PyroMark Q24 system (Qiagen Inc. USA) following the manufacturer’s instructions. *ALK* rearrangement in interphase tumor cells was investigated by fluorescence in situ hybridization (FISH) analysis using the *ALK* Break Apart FISH Probe Kit (Vysis, Abbott Park, IL, USA) whereas *ROS1* rearrangement was assessed by ZytoLigh^®^ SPEC ROS1 Break Apart Probe (ZytoVision, Bremerhaven, Germany, GmbH) [21].

“Hot spot” regions that are frequently mutated in human cancer genes were investigated by a next-generation sequencing (NGS)-based assay using the Ion AmpliSeq Cancer HotSpot Panel v2 on the Ion PGM system [22]. The genes included in this panel are *ABL1*, *AKT1*, *ALK*, *APC*, *ATM*, *BRAF*, *CDH1*, *CDKN2A*, *CSF1R*, *CTNNB1*, *EGFR*, *ERBB2*, *ERBB4*, *EZH2*, *FBXW7*, *FGFR1*, *FGFR2*, *FGFR3*, *FLT3*, *GNA11*, *GNAS*, *GNAQ*, *HNF1A*, *HRAS*, *IDH1, IDH2*, *JAK2*, *JAK3*, *KDR*, *KIT*, *KRAS*, *MET*, *MLH1*, *MPL*, *NOTCH1*, *NPM1*, *NRAS*, *PDGFRA*, *PIK3CA*, *PTEN*, *PTPN11*, *RB1*, *RET*, *SMAD4*, *SMARCB1*, *SMO*, *SRC*, *STK11*, *TP53* and *VHL*. Details on the manufacturer’s list of targeted genes and regions can be found at https://www.thermofisher.com/order/catalog/product/4475346.

Germinal DNA was extracted from peripheral blood leukocytes using the EZI DNA Blood Kit and the BioRobot EZI Workstation (Qiagen Inc.). Primer sequences for amplification and sequencing of the *CDH1* exons were based on those reported previously [23]. PCR reaction products were sequenced using the ABI BigDye Terminator Sequencing Kit (Applied Biosystems, Foster City, CA, USA) on an Applied Biosystems 3130 automated sequencer [23].

A formalin-fixed, paraffin embedded tumor block was cut into 4 μ-thick sections for H&E and immunostaining. Immunohistochemistry was performed by using the mouse monoclonal antibody against human E-cadherin (clone 36, Ventana Medical System, Tucson, AZ, USA).

## 4. Results

DNA from tumor tissue was found to be *EGFR* mutation-negative and *ALK* and *ROS1* translocation-negative. Analysis of an extensive mutation panel of genes using the Ion AmpliSeq Cancer HotSpot V2 Panel covering approximately 2800 “hotspot mutations” from 50 oncogenes and tumor suppressor genes identified a frameshift insertion c.44_45insC p.(Gln16Serfs*13) in the *TP53* gene and both a missense rs35606263 variant c.214G>A p.(Asp72Asn) and a small del/ins c.208_211delTCCCinsCCTT) in the *CDH1* gene (Table 1).

We confirmed the putative *TP53* and *CDH1* variants by one additional NGS assay using a new DNA extraction from the tumor specimen. Through this analysis it has been highlighted that the two *CDH1* variants were placed in the same chromosome (*cis*-position).

Furthermore, only the c.214G>A variant was confirmed from germline DNA by Sanger Sequencing [24] and in the tumor sample, the immunohistochemical (IHC) evaluation of the E-cadherin protein, produced by the *CDH1* gene, showed a regular expression (Figure 4).

## 5. Discussion

*EGFR* wild-type patients with BMs exhibited a significantly poorer response to *EGFR*-TKIs and a shorter survival compared with *EGFR* mutant patients [14,18]. Chemotherapy with cisplatin–pemetrexed is usually planned with only palliative intent for treatment of *EGFR* wild-type patients with BMs. It was reported a median overall survival of approximately 11 months in responder asymptomatic patients [8] whereas long-term durable remission is an exceptional event in symptomatic cases with high brain tumor burden [7,15].

Here, we report a rare outstanding clinical response to combined chemo/radiation therapy and a long survival (58 months) in a 62-years-old man with *EGFR* wild-type NSCLC patient with diffuse BMs which makes this case report particularly thought-provoking.

In order to gain better insight into this unexpectedly good clinical result, in addition to alterations in *EGFR*, *ALK* and *ROS1* genes, we also examined 2800 hotspot variants most frequently observed in oncogene and tumor suppressor genes using a NGS based assay.

Three variants, one in the *TP53* gene and two in the *CDH1* gene were found. A frameshift variant never before reported caused by the insertion c.44_45insC changing the Gln aminoacid in a Ser p.(Gln16Ser fs*13), which presumably stopped at codon 13 downstream of the insertion into the protein-building process, was detected in the *TP53* gene. It is known that about 50% of NSCLC patients harbor somatic *TP53* gene variants in the tumor mass and that the inactivation of the tumor suppressor protein p53 is a predictive factor of poor prognosis [25,26]. Despite *EGFR* wild-type and *TP53* variant observed in the tumor specimen, the patient showed an elevated sensitivity to chemo/radiation therapy.

Differences in p53 protein expression may be implicated in either the resistance or the sensitivity to chemo/-radiation therapy. In the case presented here, the *TP53* gene variant may involve lower overall cellular activity of the p53 protein that decreased DNA damage repair making the tumor cells much more unstable and prone to further genomic damages induced by chemo/radiation treatments. On the other hand, a p53 activation by acute DNA damage and hypoxia could promote the p53-mediated apoptosis of the tumor cells by unrepaired DNA damage [27]. However, further studies could be performed to examine whether the kind of anti-cancer drug used produces a superior response in tumor cells harboring *TP53* variants resulting in a complete loss of function of one allele.

Two variants in the CDH1 gene were detected. The first one consists of a missense substitution in exon 3 c.214G>A p.(Asp72Asn) that leads to the removal of a negatively charged aminoacid from the peptide chain. The variant affects the precursor region of the protein, a region that is normally cleaved to generate the mature protein. Sift and PolyPhen-2 analyses both showed that the p.Asp72Asn variant may possibly damage the protein function, while data score (by −2.162) using the Provean tool did not support a detrimental effect for this variant. Therefore, the pathogenetic role of this variant is actually of uncertain biological relevance.

The other *CDH1* variant leads to the substitution of serine and leucine with proline and phenylalanine in positions 70 and 71. This variant has not, to our knowledge, been reported before, but this substitution is considered by Provean score prediction to be deleterious (score −2.592).

The two *CDH1* variants, c.214G>A and del/ins c.208_211delTCCCinsCCTT, were shown in a cis configuration through NGS analysis performed using the tumor tissue section. Only the c.214G>A variant was detectable after Sanger analysis of DNA from the peripheral blood. These results indicated that c.214G>A represented a germline variant, while the latter, the del/ins c.208_211delTCCCinsCCTT, probably occurred as a second event in the tissue. Moreover, the normal IHC E-cadherin expression in the tumor specimen makes it unlikely that such *CDH1* variants might have a detrimental action on E-cadherin expression.

Interestingly, even though it is limited to a cancer hotspot panel, the NGS analysis performed in primary tumor sample showed only three variants, indicating that the patient could have a low tumor mutational burden (TMB). Indeed, high TMB in patients with NSCLC seems to be a poor prognostic factor and stage I NSCLC patients with high TMB did show higher recurrence rates compared with those seen in patients with lower TMB [28]. Very large radiotherapy doses per fraction lead to increasingly saturated error-free DNA damage repair with a considerably higher rate of misrepaired DNA alterations [29]. This effect, favored by a lower overall cellular activity of the p53 protein, could create, at least, an anti-tumor vaccine based on existing tumor antigens via necrotic or immunogenic cell death modulating the host inflammatory response [30]. Despite its limitations, the 2800 hotspot mutation kit could represent an useful tool as a surrogate for a more extensive full genome analysis in the selection of patients more or less sensitive to chemotherapy or immunotherapy.

In conclusion, we cannot exclude that other molecular factors could be involved and contribute in the determination of the long-term survival of this patient. However, this case report suggests that high dose of radiotherapy associated with chemotherapy may be effective in patients where a lower TMB and a defective DNA repair efficiency is detected.

## Figures and Tables

**Figure 1 diagnostics-09-00042-f001:**
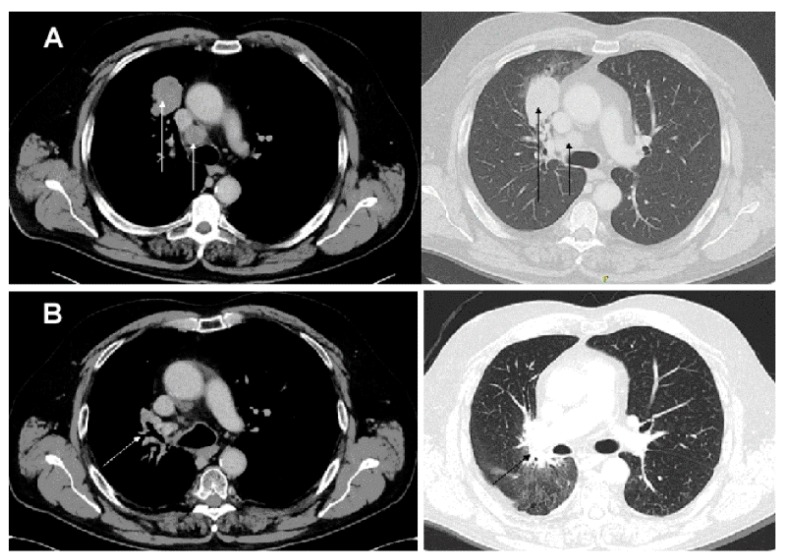
Computerized tomography scan performed at initial diagnosis: (**A**) tumor mass in the right upper lung’s lobe with omolateral mediastinal lymphadenopathy on arrows; (**B**) complete response of the primary lung lesion after the intensity-modulated radiation therapy (IMRT).

**Figure 2 diagnostics-09-00042-f002:**
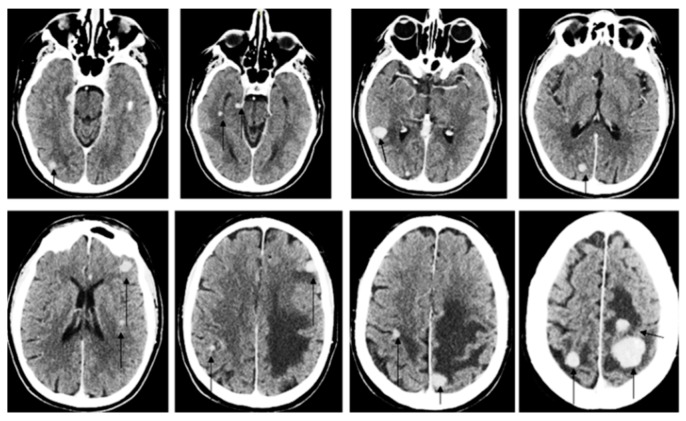
Computerized tomography scan performed at initial diagnosis shows multiple brain metastases (BMs) indicated by arrows.

**Figure 3 diagnostics-09-00042-f003:**
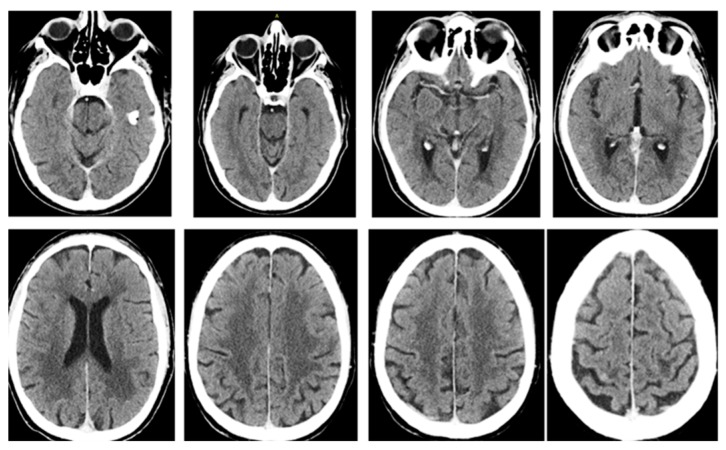
Computerized tomography scan shows complete response of multiple BMs after intensity-modulated radiation therapy (IMRT).

**Figure 4 diagnostics-09-00042-f004:**
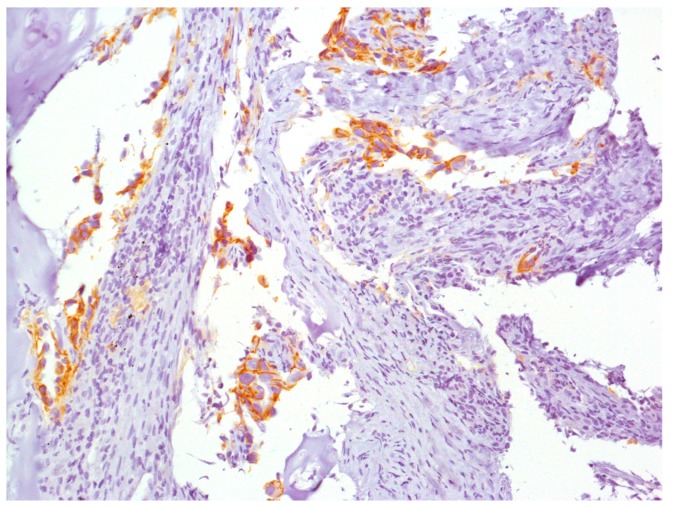
Immunohistochemical staining of the bronchial biopsy sample (4×). Aggregates of roughly oval or elongated neoplastic cells with evident nuclei infiltrating the bronchial stroma can be seen along with extensive cytoplasmic expression of E-cadherin.

**Table 1 diagnostics-09-00042-t001:** Gene variants identified by NGS analysis.

# Locus	Gene	Exon	c.DNA Change	Protein Change	Coverage	% Allelic Variant	Function
chr17: 7579868	*TP53*	2	c.44_45 insC	p.Gln16Ser fs*13	1834	18.1	Frameshift Insertion
chr16: 68835623	*CDH1*	3	c.214G>A (rs35606263)	p.Asp72Asn	420	44.7	missense
chr16: 68835615	*CDH1*	3	c.208_211delTCCCinsCCTT	p.Ser70_Leu71delins ProPhe	343	47.7	deletion/insertion

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
