# Peer review of "Complete and Durable Response to Combined Chemo/Radiation Therapy in EGFR Wild-Type Lung Adenocarcinoma with Diffuse Brain Metastases"

_diagnostics, 2019, doi:10.3390/diagnostics9020042_

Round 1
Reviewer 1 Report
The manuscript “Complete and Durable Response to Combined Chemo/Radiation Therapy in EGFR Wild-Type Lung Adenocarcinoma with Diffuse Brain Metastases” by Santeufemia et al. describes a case-report of the treatment of a non-smoker male patient with multiple, large and diffuse brain metastases from EGFR wild-type lung adenocarcinoma with a combined, highly aggressive chemo/radiation therapy. The outcome was the complete remission of the disease. The mutational analysis of a panel of oncogenes and tumor suppressor genes shows mutations in TP53 and CDH1 genes only. The authors hypothesize the reasons why these mutations could explain the treatment response.
Although the manuscript does not provide the molecular characteristics that explain the excellent response of the patient to the treatment, it may be interesting as case-report.
Comments:
1. The explanation of the treatment underwent by the patient deserve a better explanation. It should be compared in detailed with the common treatment received by other wild-type EGFR symptomatic NSCLC patients. It should be clearly explained why the treatment for this specific patient is classified as “highly aggressive” compared to others received by similar patients.
2. Introduction section is very short, it should include more information about the standard treatment approach in this specific clinical situation. Moreover, Discussion section should mention similar studies (e.g. case-report studies or clinical studies) to treat this kind of patients.
3. Lines 52-53: “Unfortunately, EGFR wild-type patients with BMs exhibited a significantly poorer response and a shorter survival compared with EGFR mutant patients” It should be mentioned/explained the response to WHAT TREATMENT the authors refer to. Do they refer to TKIs? It should be explained that the TKIs mentioned (erlotinib, gefitinib and afatinib) are tyrosine kinase inhibitors specific for epidermal growth factor receptor (EGFR TKIs).
4. Figure 4 needs a better explanation at the figure legend.
5. Lines 175-177: “Despite the NGS analysis found c.214G>A and del/ins c.208_211delTCCCinsCCTT in a cis configuration, only the c.214G>A was confirmed at germinal level by Sanger analysis, suggesting its somatic origin.” Please, rephrase the sentence to clarify which mutation is somatic and which is germinal.
6. Lines 110-117: Describes the analysis of E-cadherin (both at the gene (CDH1) and protein levels (by immunohistochemistry)). However, E-cadherin has not been mentioned before. Information about E-Cadherin and its important role in tumors should be included before (e.g. Introduction section).
7. Line 47: “Epidermal Growth Factor (EGFR) gene” should be Epidermal Growth Factor Receptor (EGFR) gene.
8. Lines 111-112: “Primer sequences for the amplification and sequence of the CDH1 exons were based on those reported previously”: The reference is missing.
Author Response
We thanks the reviewer for her/his interesting considerations and for considering the case report suitable for publication in diagnostics. Thanks to the useful suggestions the manuscript has been notably improved. Below our point-to-point answers and the action taken.
1.The explanation of the treatment underwent by the patient deserve a better explanation. It should be compared in detailed with the common treatment received by other wild-type EGFR symptomatic NSCLC patients. It should be clearly explained why the treatment for this specific patient is classified as “highly aggressive” compared to others received by similar patients.
Reply We are agree with the reviewer that the chemotherapy treatment must be better explained. Moreover, we specified the reason because we considered highly aggressive the treatment performed in our patient and we compared it with that actually considered as the standard chemotherapy regimen.
Lines 83-90
Indeed, in addition to the WBRT and four cycles of cisplatin-pemetrexed combination, considered the standard treatment [12], we decided to administer two more cycles of the same induction chemotherapy followed by 18 pemetrexed maintenance cycles. Moreover, taking in account the excellent response to previous chemo/radiation therapy and his good performance status on the remaining lung lesion, mediastinal lymph nodes, and residual brain metastasis, a treatment consisting of intensity-modulated radiation therapy (IMRT) and helical tomotherapy (HT) has been performed
2. Introduction section is very short, it should include more information about the standard treatment approach in this specific clinical situation. Moreover, Discussion section should mention similar studies (e.g. case-report studies or clinical studies) to treat this kind of patients.
Reply. As indicated by the reviewer the introduction has been implemented including more information about the standard treatment approach. Moreover other similar studies have been reported in the discussion session.
Lines: 46-63
Treatment via whole brain radiotherapy (WBRT) has shown to improve survival time by approximately 4 months after diagnosis and it represents the standard palliative treatment for NSCLC symptomatic patients with multiple BMs who are not candidates to receive either surgery or radiosurgery [4-6]. Although the role of chemotherapy remains unclear in this setting, as most chemotherapeutic drugs are not able to cross the blood-brain barrier (BBB) [7], it is thought that a cisplatin-pemetrexed combination may be a viable option in the treatment of asymptomatic Epidermal Growth Factor Receptor (EGFR) wild-type non squamous NSCLC patients with BMs [8]. Indeed, it is known that the BBB is disrupted when BMs develop and, in addition, the cerebrospinal fluid penetration of intravenous pemetrexed in patients with BMs has been previously demonstrated [9-10].
Moreover, the administration of four cycles of cisplatin plus pemetrexed, followed, in absence of disease progression, by maintenance of pemetrexed alone, still represents the standard chemotherapy regimen adopted in Italy in fit patients affected by EGFR wild-type advanced pulmonary adenocarcinoma without programmed death-ligand 1 (PD-L1) immunohistochemical expression on ≥50% of tumor cells [11-12].
Lines:75-79
Chemotherapy with cisplatin–pemetrexed is usually planned with only palliative intent for treating EGFR wild-type patients with BMs. In this setting it was reported a median overall survival of approximately 11 months in responder asymptomatic patients [8] whereas long-term durable remission is an exceptional event in symptomatic cases with high brain tumor burden [7].
Line 200 in the discussion section we reported other two references [7-15] useful to clarify the clinical course of these patients.
3 “Unfortunately, EGFR wild-type patients with BMs exhibited a significantly poorer response and a shorter survival compared with EGFR mutant patients” It should be mentioned/explained the response to WHAT TREATMENT the authors refer to. Do they refer to TKIs? It should be explained that the TKIs mentioned (erlotinib, gefitinib and afatinib) are tyrosine kinase inhibitors specific for epidermal growth factor receptor (EGFR TKIs).
Reply. We recognize that in the text it is not clearly mentioned/explained to what treatment the EGFR mutant patients are submitted.. Now we better specified that the treatment performed in this setting is TKi based. Moreover we explained that the tyrosine kinase inhibitors are drugs specific for EGFR.
Lines: 64-68
On the other hand BMs incidence in EGFR mutant NSCLC is higher than in wild-type EGFR disease [13]. NSCLC patients with activating somatic mutations in the tyrosine kinase domain of EGFR gene are highly responsive to tyrosine kinase inhibitors (TKIs) such as erlotinib, gefitinib and afatinib which are specific for EGFR (EGFR-TKIs) [13-15].
4. Figure 4 needs a better explanation at the figure legend
Reply: The legend to the figure 4 have been improved following the suggestions of the reviewer.
Line:190-192
Figure 4. Immunohistochemical staining of the bronchial biopsy sample. Aggregates of roughly oval or elongated neoplastic cells with evident nuclei infiltrating the bronchial stroma, and showing extensive cytoplasmic expression of E-cadherin.
5. Lines 175-177: “Despite the NGS analysis found c.214G>A and del/ins c.208_211delTCCCinsCCTT in a cis configuration, only the c.214G>A was confirmed at germinal level by Sanger analysis, suggesting its somatic origin.” Please, rephrase the sentence to clarify which mutation is somatic and which is germinal.
Reply: as suggested by reviewer we clarified which mutation is somatic and which germinal rephrasing the sentence as below indicated.
Lines 241-246
The two CDH1 variants, c.214G>A and del/ins c.208_211delTCCCinsCCTT, were shown in a cis configuration through NGS analysis performed using the tumor tissue section. Only the c.214G>A variant was detectable after Sanger analysis of DNA from the peripheral blood. These results indicated that c.214G>A represented a germline variant, while the latter, the del/ins c.208_211delTCCCinsCCTT, probably occurred as a second event in the tissue.
6. Lines 110-117: Describes the analysis of E-cadherin (both at the gene (CDH1) and protein levels (by immunohistochemistry)). However, E-cadherin has not been mentioned before. Information about E-Cadherin and its important role in tumors should be included before (e.g. Introduction section).
Reply: as suggested by reviewer in introduction section we have added a paragraph describing the important role of TP53 and CDH1 genes in tumors.
Lines: 93-102
Specifically, we focused our attention on TP53 and CDH1 genes which we found to be altered in the tissue sample. TP53 represents an important tumor suppressor gene, widely known to contribute to oncogenesis of several neoplastic diseases, including lung cancer [19], through transcriptional regulation of numerous genes involved in cell death, cell cycle arrest, senescence, DNA-repair and many other processes. By contrast, CDH1 gene codifies for a transmembrane glycoprotein, also known as epithelial cadherin (E-cadherin) that plays an essential role in maintaining cell adhesion and adherent junctions in normal tissues and it is frequently absent in a variety of epithelial tumors, including lung cancer [20], thus promoting cancer invasion and metastasis.
7. Line 47: “Epidermal Growth Factor (EGFR) gene” should be Epidermal Growth Factor Receptor (EGFR) gene.
Reply: as suggested by the reviewer the EGFR acronym has been rewritten as Epidermal Growth Factor ReceptorLine:54
8. Lines 111-112: “Primer sequences for the amplification and sequence of the CDH1 exons were based on those reported previously”: The reference is missing.
Reply: as suggested by the reviewer we have added the specific reference. Line:165 reference 23
Reviewer 2 Report
In the revised manuscript entitled “Complete And Durable Response To Combined Chemo/Radiation Therapy In Egfr Wild-Type Lung Adenocarcinoma With Diffuse Brain Metastases” by Santeufemia D.A., the authors presented a very interesting case report of a NSCLC patient with multiple brain metastases who underwent to highly aggressive chemo/radiation therapy with a long survival up to 58 months from diagnosis of primary tumor. The chemotherapeutic approach consisted in a combination of cisplatin and pemetrexed, as the primary tumor was EGFR wild type.
The authors performed mutational analysis by using the Ion AmpliSeq Cancer HotSpot V2 Panel covering approximately 2800 “hotspot mutations” from 50 oncogenes and 122 tumor suppressor genes. By this approach they identified a frameshift in the TP53 gene, never characterized before, and both a missense variant and a small del/ins in the CDH1 gene, indicating that the patient could have a low TMB.
This case report suggests that high dose of radiotherapy associated with chemotherapy may be effective in patients where a lower TMB is detected.
Moreover, these data suggests the possible application of the 2800 hotspot mutations kit to identify patients more or less sensible to treatments and paves the way for the characterization of the novel identified mutation variants.
In the opinion of this reviewer the data are well presented and are satisfactory for the publication in Diagnostics.
Author Response
We thanks the reviewer for her/his interesting considerations and for considering the case report suitable for publication in diagnostics.
Round 2
Reviewer 1 Report
The authors have satisfactorily addressed all my comments.
This manuscript is a resubmission of an earlier submission. The following is a list of the peer review reports and author responses from that submission.